# UIR-LoRA: Achieving Universal Image Restoration through Multiple Low-Rank Adaptation

## Abstract

Existing unified methods typically treat multi-degradation image restoration as a multi-task learning problem. Despite performing effectively compared to single degradation restoration methods, they overlook the utilization of commonalities and specificities within multi-task restoration, thereby impeding the model's performance. Inspired by the success of deep generative models and fine-tuning techniques, we proposed a universal image restoration framework based on multiple low-rank adapters (LoRA) from multi-domain transfer learning. Our framework leverages the pre-trained generative model as the shared component for multi-degradation restoration and transfers it to specific degradation image restoration tasks using low-rank adaptation. Additionally, we introduce a LoRA composing strategy based on the degradation similarity, which adaptively combines trained LoRAs and enables our model to be applicable for mixed degradation restoration. Extensive experiments on multiple and mixed degradations demonstrate that the proposed universal image restoration method not only achieves higher fidelity and perceptual image quality but also has better generalization ability than other unified image restoration models.

## 1 Introduction

In the wild, a range of distortions commonly appear in captured images, including noise[56], blur[14, 47, 6], low light[58, 22, 8], and various weather degradations[15, 51, 54, 45]. As a fundamental task in low-level vision, image restoration aims to eliminate these distortions and recover sharp details and original scene information from corrupted images. With the assistance of deep learning, an abundance of restoration approaches [56, 3, 54, 2, 16, 14, 53] have made significant progress in eliminating single degradation from images. However, these approaches typically require additional training from scratch on specific image pairs in multi-degraded scenarios, which leads to inconvenience in usage and limited generalization ability.

For simplicity and practicality, some existing works [15, 31, 55]consider training a unified model (also called all-in-one model) to handle multiple degradations as multi-task learning. These studies primarily explore how to discern degradation from the image and integrate it into the restoration network. Nevertheless, these methods share all parameters across different degradations, resulting in gradient conflicts [40, 52] that hinder further improvement of unified models' performance.

Digging deeper, the underlying issue lies in that the similarities among different image restoration tasks and the inherent specificity of each degradation are not well considered and utilized in the training. This limitation drives us to seek solutions for multi-degradation restoration by leveraging both commonalities and specificities.

Submitted to 38th Conference on Neural Information Processing Systems (NeurIPS 2024). Do not distribute.

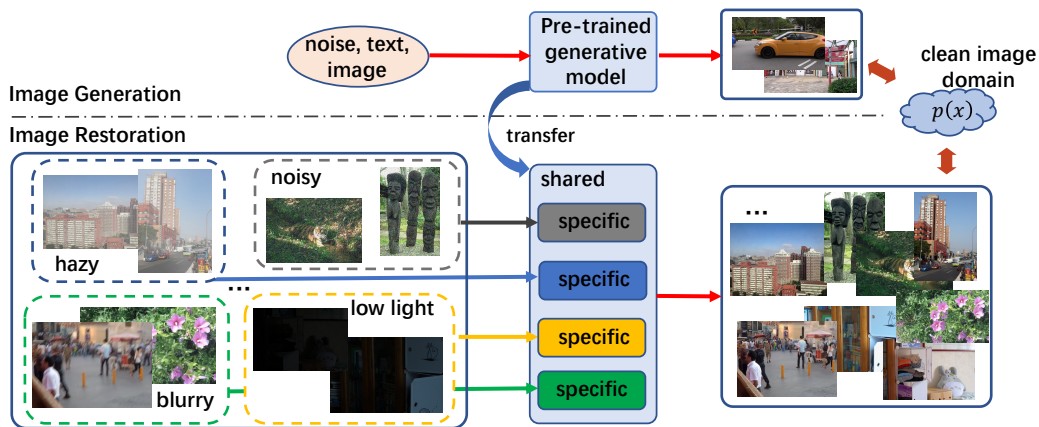

Figure 1: Motivation of our work. A pre-trained generative model serves as the shared component and minimal parameters are added to model the specificity of each degradation restoration task.

Inspired by the successes of deep generative models[37, 36, 35] and fine-tuning techniques[11, 10, 4], we propose addressing the aforementioned issue from the perspective of multi-domain transfer learning, as presented in Figure 1. The pre-trained generative model exhibits powerful capabilities, implying rich prior knowledge of clear image distribution $p(x)$, which is exactly what is needed for image restoration. Since image prior $p(x)$ is degradation-agnostic and applicable to all types of degraded images, the pre-trained generative model is an excellent candidate for serving as the shared component for multiple degradation restoration. To model the transition from the clean image domain to different degraded image domains, minimal specific parameters are required to fine-tune the pre-trained model for each degradation restoration task. This approach not only isolates conflicts between each degradation task but also ensures efficiency and performance during training.

Following the idea of multi-domain transfer learning, we proposed a universal image restoration framework based on multiple low-rank adaptations, named UIR-LoRA. In our framework, the pre-trained SD-turbo [39] serves as the shared fundamental model for multiple degradation restoration tasks due to its powerful one-step generation capability and extensive image priors. Subsequently, we incorporate the low-rank adaptation (LoRA) technique [11] to fine-tune the base model for each specific image restoration task. This involves augmenting low-dimensional parameter matrices on selected layers within the base model, ensuring efficient fine-tuning while maintaining independence between LoRAs for each specific degradation. Additionally, we propose a LoRA composition strategy based on degradation similarity. We calculate the similarity between degradation features extracted from degraded images and existing degradation types, utilizing it as weights for combining different LoRA experts. This strategy enables our method to be applicable for restoring mixed degradation images. Moreover, we conducted extensive experiments and compared our approach with several existing unified image restoration methods. The experimental results demonstrate that our method achieves superior performance in the restoration of various degradations and mixed degradations. Not only does our approach outperform existing methods in terms of distortion and perceptual metrics, but it also exhibits significant improvements in visual quality.

Our contributions can be summarized as follows:

- From the perspective of multi-domain transfer learning, we propose a novel universal image restoration framework based on multiple low-rank adaptations. It leverages the pre-trained generative model as the shared component for multi-degradation restoration and employs distinct LoRAs for multiple degradations to efficiently transfer to specific degradation restoration tasks.

- We introduce a LoRAs composition strategy based on the degradation similarity, which adaptively combines trained LoRAs and enables our model to be applicable for mixed degradation restoration.

- Through extensive experiments on multiple and mixed degradations, we demonstrate that the proposed universal image restoration method not only achieves higher fidelity and perceptual image quality but also has better generalization ability than other unified models.

## 2 Related Work

### 2.1 Image Restoration

**Specific Degradation Restoration.** According to degradation type, image restoration tasks are categorized into different groups, including denoising, deblurring, inpainting, draining *.etc*. Most existing image restoration methods [2, 53, 16, 56, 5, 14] mainly address the issue with a single degradation. Traditional approaches [27, 28, 7] have proposed image priors. While these priors can be applied to different degraded images, their capability is limited. Due to the remarkable capability of the deep neural network (DNN), numerous DNN-based methods [2, 53, 16] have been proposed to tackle image restoration tasks. While DNN-based methods have made significant progress, they struggle with multiple degradations and mixed degradations, since they typically require retraining from scratch on data with the same degradation.

**Universal degradation restoration.** Increasing attention is currently focused on developing a unified model to process multiple degradations. For example, AirNet[15] explores the degradation representation in latent space for separating them in the restoration network. PromptIR[31] utilizes a prompt block to extract the degradation-related features to improve the performance. Daclip-IR[20] introduces the clip-based encoder to distinguish the type of degradation and extract the semantics information from distorted images and embed them into a diffusion model to generate high-quality images. Despite the advancements, these unified models still have limitations. They also require retraining all parameters when unseen degradations arrive and have limited performance due to the gradient conflict.

### 2.2 Low-Rank Adaptation

LoRA [11] is proposed to fine-tune large models by freezing the pre-trained weights and introducing trainable low-rank matrices. This fine-tuning method leverages the property of "intrinsic dimension" within neural networks, lowering the rank of additional matrices and making the re-training process efficient. Concretely, given a weight matrices $W \in \mathbb{R}^{n \times m}$ in pre-trained model $\theta_p$, two trainable matrices $B \in \mathbb{R}^{n \times r}$ and $A \in \mathbb{R}^{r \times m}$ are inserted into the layer to represent the LoRA $\Delta W = BA$, where $r$ is the rank and satisfy $r \ll mim(n, m)$, the updated weights $W'$ are calculated by

$$W' = W + \Delta W. \tag{1}$$

By applying LoRA in pre-trained models, numerous image generation methods [29, 13], show superior performance in the field of image style and semantics concept transferring. Additionally, fine-tuning methods like ControlNet [57], T2i-adapter [24] are also commonly employed in large-scale pre-trained generative models such as Stable Diffusion [37], SDXL [30], and Imagen [38].

### 2.3 Mixture of Experts

Mixture of Experts (MoE) [41, 49, 48] is an effective approach to scale up neural network capacity to improve performance. Specifically, MoE integrates multiple feed-forward networks into a transformer block, where each feed-forward network is regarded as an expert. A gating function is introduced to model the probability distribution across all experts in the MoE layer. The gating function is trainable and determines the activation of specific experts within the MoE layer based on top-k values. Broadly speaking, our framework aligns with the concept of MoE. However, unlike traditional MoE layers, we employ the more efficient LoRA as experts in selected frozen layers and utilize a degradation-aware router across all selected layers to uniformly activate experts, reducing learning complexity and avoiding conflicts among different image restoration tasks on experts.

## 3 Methodology

### 3.1 Problem Definition

This paper seeks to develop a novel universal image restoration framework capable of handling diverse forms of image degradation in the wild by fine-tuning the pre-trained generative model. Consider a set of $T$ image restoration tasks $D = \{D^k\}_{k=1}^T$, where $D^k = \{(x_i, y_i)\}_{i=1}^{n_k}$ is the training dataset containing $n_k$ images pairs of the $k$-th image degradation task. Within the set of tasks $D$,

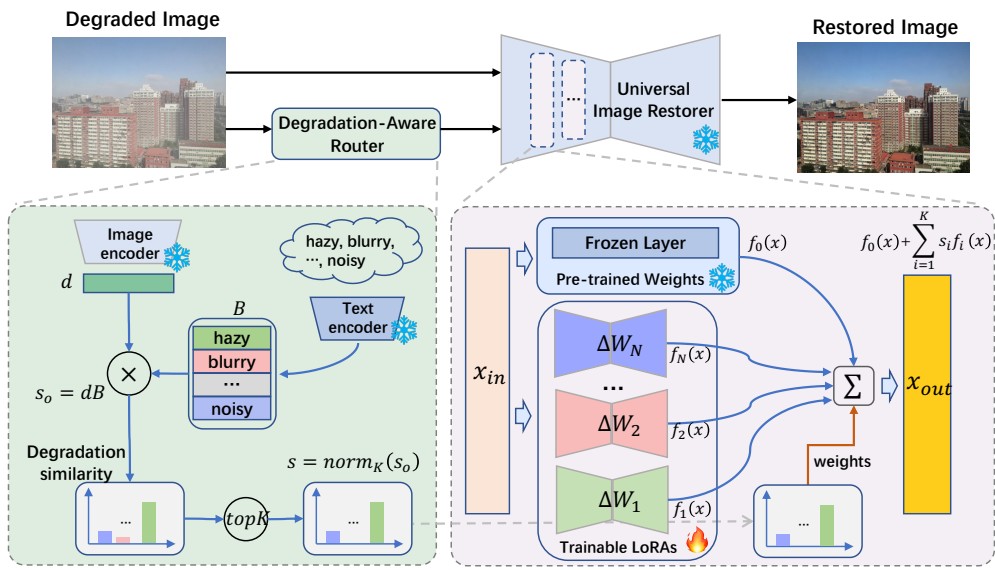

Figure 2: Overview of UIR-LoRA. UIR-LoRA consists of two components: a degradation-aware router and a universal image restorer. The router calculates degradation similarity in the latent space of CLIP, while the restorer utilizes the similarity provided by the router to combine LoRAs and frozen base model and restore images with multiple or mixed degadations.

each task $D^k$ only has a specific type of image degradation, with no intersection between any two tasks. Given a pre-trained generative model $\theta_p$ with frozen parameters, our objective is to learn a set of composite $\{\theta_k\}_{k=1}^{T}$ to construct a unified model $f_\theta$ that performs well on multi-degradation restoration and mixed degradation restoration by transferring learning, where $\theta = \theta_p + \sum_{k=1}^{T} s_k \theta_k$ and $s_k$ represents the composite weight for $\theta_k$. The trainable $\{\theta_k\}_{k=1}^{T}$ can be optimized through minimizing the overall image reconstruction loss:

$$L = E_{(x,y) \in D} l(f_\theta(x), y). \tag{2}$$

We will present how to design and optimize the trainable $\{\theta_k\}_{k=1}^{T}$ and construct the composite weights $s$ in the next sections.

## 3.2 Overview of Universal Framework

Inspired by transferring learning, we introduce a novel universal image restoration framework based on multiple low-rank adaptations, named UIR-LoRA. Referring to Figure 2, our framework consists of two main components, namely degradation-aware router and universal image restorer, respectively. The degradation-aware router first extracts the degradation feature from input degraded images and then calculates the similarity probabilities $s$ with existing degradations in the latent space of CLIP model [35, 20]. For the universal image restorer, it comprises a pre-trained generative model $\theta_p$ and $T$ trainable LoRAs $\{\theta_k\}_{k=1}^{T}$. This design is primarily motivated by two considerations: firstly, the pre-trained generative model contains extensive image priors that are degradation-agnostic and can be shared across all types of degraded images. Secondly, each LoRA can independently capture specific characteristics of each degradation without gradient conflicts. In practice, the pre-trained SD-turbo [39] is employed as the frozen base model in our framework and each LoRA $\theta_k$ serves as an expert responsible for transferring the frozen base model to a specific degradation restoration task $D^k$. By adjusting the value of Top-K parameter within the degradation-aware router, different combinations of LoRAs in the universal image restorer can be activated, enabling the removal of a specific degradation and mixed degradation in multi-degraded scenarios.

## 3.3 Degradation-Aware Router

The Degradation-Aware Router is designed to provide the restorer with weights for LoRA combination based on degradation confidence. Following Daclip-ir [20], we utilized the pre-trained image encoder in CLIP [35] to obtain the degradation vector $d \in \mathbb{R}^{1 \times z}$ from the input degraded image $x$, where $z$ is degradation length in latent space. Differing from Daclip-ir [20], we use the degradation vector and existing degradations to calculate the similarity, instead of directly embedding the degradation vector into the restoration network in Daclip-ir [20]. The existing degradations refer to the vocabulary bank of diverse degradation types that we introduce in the router, such as "noisy", "blurry" and "shadowed". This vocabulary bank is highly compact and flexible when adding new degradation types. Similarly, by applying the text encoder of CLIP [35], the vocabulary bank can be encoded into the degradation bank $B \in \mathbb{R}^{z \times T}$ in the latent space. As presented in Figure 2, the original degradation similarity $s_o \in \mathbb{R}^{1 \times T}$ is calculated by:

$$s_o = dB. \tag{3}$$

Building upon the original similarity, we adopt a more flexible and controllable Top-K strategy to modify $s_o$. Specifically, we select the Top-K largest values from the original similarity $s_o$, and normalize them to reallocate the weights for LoRAs. The reallocation process can be formulated as :

$$s = \frac{s_o \cdot M_K}{\sum s_o \cdot M_K}, \tag{4}$$

where $M_K$ represents a binary mask with the same length as $s_o$, where it is 1 when the corresponding value in $s_o$ is among the Top-K, otherwise it is 0. With a smaller value of $K$, the restorer activates fewer LoRAs, reducing its computational load. For instance, with $K = 1$, only the most similar LoRA is activated and it yields effective results when $s$ is accurate, but performance noticeably declines with inaccurate $s$. Conversely, as $K$ increases, the restorer exhibits higher tolerance to $s$ and the combination of LoRAs allows it to handle mixed degradation.

## 3.4 Universal Image Restorer

Our universal image restorer consists of a pre-trained generative model $\theta_p$ and a set of LoRAs $\{\theta_k\}_{k=1}^T$. As illustrated in Figure 2, our universal image restorer takes the degraded image $x$ and similarity $s$ predicted by the degradation-aware router as inputs. It then activates relevant LoRAs based on $s$ to recover the degraded image along with the frozen base model. Since one of our objectives is to ensure that each LoRA serves as an expert in processing a specific degradation, the number of LoRAs in the restorer aligns with the number of degradation types, $T$. In practice, we select multiple layers from the base model, For a selected layer $W$ of the pre-trained base model, a sequence of trainable matrices $\{\Delta W_k\}_{k=1}^T$ are added into this layer, and the parameters of all chosen layers $L$ form a complete LoRA $\theta_k = \{\Delta W_k^j\}_{j \in L}$. As previously explained, each LoRA is a unique expert responsible for a specific degradation. Drawing inspiration from Mixture of Expert (MoE), we aggregate the outputs of each expert rather than directly merging parameters in [11]. Therefore, given the input feature $x_{in}$ of the current layer and the similarity $s$, the total output $x_{out}$ of this modified layer can be expressed as

$$x_{out} = f_o(x_{in}) + \sum_{i=1}^{K} s_i f_i(x_{in}), \tag{5}$$

where $f_i(x_{in})$ denotes the result of $i$-th trainable matrice $W_i$, particularly $f_o(x_{in})$ is output of the frozen base layer. From the equation 5, it can be observed that the introduced LoRAs interact with the frozen base model at intermediate feature layers in our restorer. This interaction forces the restorer to leverage the image priors of the pre-trained generative model and eliminate degradation with the assistance of LoRAs. In contrast to employing stable diffusion [37] directly as a post-processing technique, our restorer yields results closer to the true scene without introducing inaccurate structural details. Since each $W$ is implemented using two low-rank matrices like the formula 1, the total trainable parameters of our framework are much smaller than that of the pre-train generative model.

## 3.5 Training and Inference Procedure

During the training phase, for the efficient training of the universal image restorer, we ensure that each batch is sampled from the same degradation type $D^k$, and activate the corresponding LoRA $\theta^k$

Table 1: Comparison of the restoration results over ten different datasets. The best results are marked in boldface.

| Model | Distortion | | Perceptual | | Complexity | |
|---|---|---|---|---|---|---|
| | PSNR↑ | SSIM↑ | LPIPS↓ | FID↓ | Param /M | Runtime /s |
| SwinIR [16] | 23.37 | 0.731 | 0.354 | 104.37 | 15.8 | 0.66 |
| NAFNet [2] | 26.34 | 0.847 | 0.159 | 55.68 | 67.9 | 0.54 |
| Restormer [53] | 26.43 | 0.850 | 0.157 | 54.03 | 26.1 | 0.14 |
| AirNet [15] | 25.62 | 0.844 | 0.182 | 64.86 | 7.6 | 1.50 |
| PromptIR [31] | 27.14 | 0.859 | 0.147 | 48.26 | 35.6 | 1.19 |
| IR-SDE [21] | 23.64 | 0.754 | 0.167 | 49.18 | 36.2 | 5.07 |
| DiffBIR [17] | 21.01 | 0.618 | 0.263 | 91.03 | 363.2 | 5.95 |
| Daclip-IR [20] | 27.01 | 0.794 | 0.127 | 34.89 | 295.2 | 4.09 |
| **UIR-LoRA (Ours)** | **28.08** | **0.864** | **0.104** | **30.58** | 95.2 | 0.44 |

for training. Since the dataset $D$ is organized by degradation type without overlap and each LoRA is assigned to handle each type of degradation correspondingly, the overall optimization process in equation 2 can be decomposed into independent optimization processes for each degradation. This design and training process circumvent task conflicts among multiple degradations and makes it possible to use suitable loss functions for the specific degradation. Due to the availability of accurate $s$ during training and the use of pre-trained encoders from CLIP [35] and Daclip-ir [20] in our router, the router was not utilized during training.

In the inference phase, the similarity $s$ is unknown and needs to be estimated from the degraded image. The estimated similarity $s$ serves as a reference in our framework and can also be manually specified by users. Subsequently, our universal image restorer composite LoRAs and recovers the input image with the guidance of $s$.

## 4 Experiments

### 4.1 Experimental Setting

**Datasets.** We validate the effectiveness of our framework in multiple and mixed degradation scenarios. In the case of multiple degradations, we follow Daclip-IR [20] and construct a dataset using 10 different single degradation datasets. Briefly, the composite dataset comprises a total of 52800 image pairs for training and 2490 image pairs for testing. The degradation types included are commonly encountered in image restoration, such as blur, noise, shadow, JPEG compression, and weather degradations. For mixed degradations, we utilize two degradation datasets, REDS [25] and LOLBlur [58]. In REDS, the images are distorted by JPEG compression and blur, and those images in LOLBlur have blur and low light. For more details about datasets in our experiments, please refer to **Appendix**.

**Metrics.** The objective of the image restoration task is to output images with enhanced visual quality while maintaining high fidelity to the original scene information. This differs from image generation tasks, which prioritize visual quality. Therefore, to thoroughly evaluate the effectiveness of our method, we utilize reference-based image quality assessment techniques from both distortion and perceptual perspectives, including PSNR, SSIM, and LPIPS, as well as FID.

**Comparison Methods.** In the experiments, we primarily compare with several state-of-the-art methods in image restoration, which fall into two categories: regression model and generative model. Regression models include NAFNet [2], Restormer [53], as well as AirNet [15] and PromptIR [31] proposed for multiple degradation restoration. DiffBIR [17], IR-SDE [21] and Daclip-IR [20] are generative models built upon the diffusion model [9].

### 4.2 Implementation Details

During the training, we adapt an AdamW optimizer to update the weights of trainable parameters in our model. Before training LoRA for specific degradation, we add skip-connections in the VAE of SD-turbo[39] like [29, 44] and train them with multiple degraded images. We set the initialization

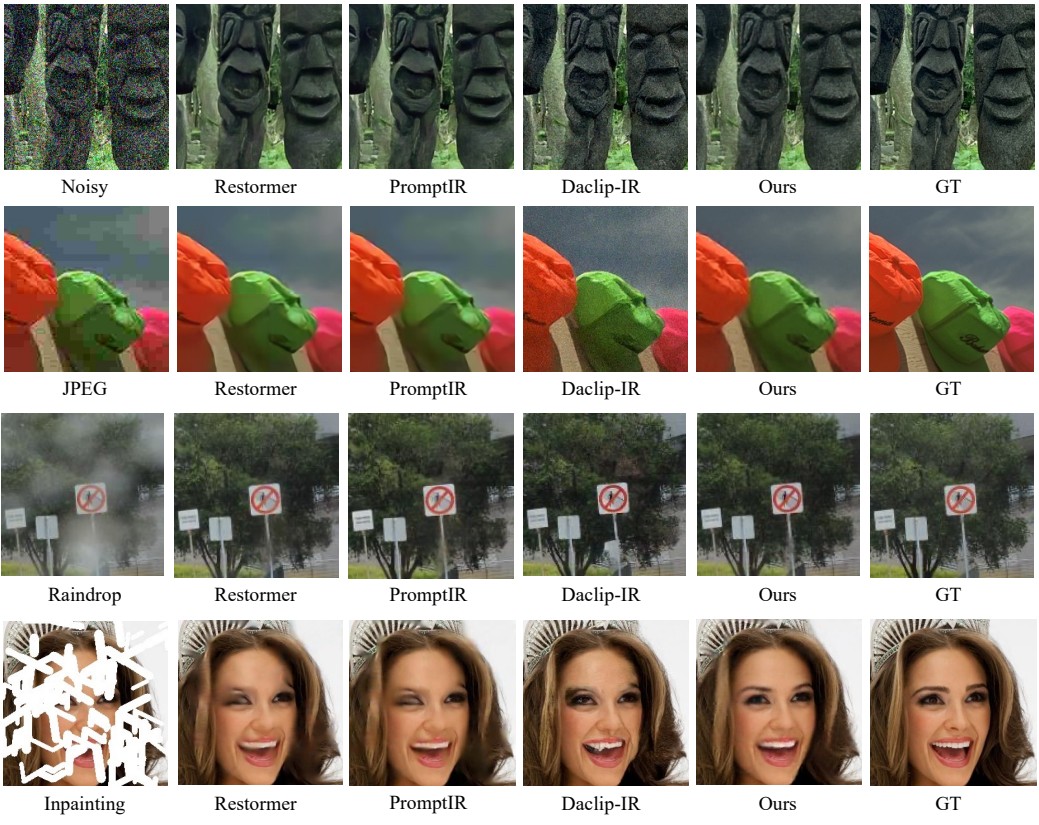

Figure 3: Qualitative comparison on multiple degraded images.

learning rate to 2e-4 and decrease it with CosineAnnealingLR . We trained every LoRA for 80K iterations with batch size 8 and we keep the same hyper-parameters when training different LoRAs. The default rank of LoRAs in VAE and Unet is 4 and 8, respectively.

## 4.3 Multiple Image Restoration

For fair comparisons, all methods are trained and tested on the multiple degradation dataset. The results are presented in Table 1. We can find that our model, UIR-LoRA, considerably surpasses all compared image restoration approaches across four metrics. This indicates that our approach can balance generating clear structures and details while ensuring the restored images closely resemble the original information of the scene. The visual comparison results depicted in Figure 7 also confirm this assertion. Regression models such as NAFNet [2]and Restormer [53], lacking extensive image priors, tend to produce blurred and over-smoothed images, leading to inferior visual outcomes. Conversely, generative models Daclip-IR [20] excessively prioritize perceptual quality, yielding artifacts and noise that diverge from the actual scene information. Our approach integrates the strengths of both categories of methods, enabling strong performance in both distortion and perceptual aspects

## 4.4 Mixed Image Restoration

To evaluate the transferability of UIR-LoRA, we conduct some experiments on mixed degradation datasets from REDS[25] and LOLBlur [58]. Each image in these two datasets contains more than one type of degradation, like blur, jpeg compression, noise, and low light. We test the mixed degraded images using models trained on multiple degradations and set $K$ to 2 in the router. As shown in Table 2, our method achieves superior results in both distortion and perceptual quality, particularly on the LOLBlur dataset. We also provide visual comparison results, as illustrated in Figure 4, our approach effectively enhances the low-light image compared to SOTA methods, highlighting its stronger transferability in the wild. More visual results can be found in **Appendix**.

Table 2: Comparison of the restoration results on mixed degradation datasets. The best results are marked in boldface.

| Model | REDS | | | | LOLBlur | | | |
|---|---|---|---|---|---|---|---|---|
| | PSNR↑ | SSIM↑ | LPIPS↓ | FID↓ | PSNR↑ | SSIM↑ | LPIPS↓ | FID↓ |
| SwinIR | 21.53 | 0.676 | 0.449 | 116.80 | 10.06 | 0.320 | 0.619 | 124.52 |
| NAFNet | 25.06 | **0.721** | 0.412 | 122.12 | 10.57 | 0.397 | 0.477 | 85.77 |
| Restormer | 23.15 | 0.713 | 0.413 | 118.61 | 12.77 | 0.479 | 0.478 | 87.23 |
| PromptIR | 24.98 | 0.712 | 0.424 | 128.11 | 9.09 | 0.275 | 0.560 | 91.68 |
| DiffBIR | 20.70 | 0.598 | 0.377 | 122.76 | 9.86 | 0.288 | 0.611 | 125.41 |
| Daclip-IR | 24.30 | 0.699 | 0.337 | 95.29 | 14.52 | 0.599 | 0.358 | 68.10 |
| **UIR-LoRA** | **25.11** | 0.718 | **0.315** | **89.79** | **18.16** | **0.690** | **0.318** | **61.55** |

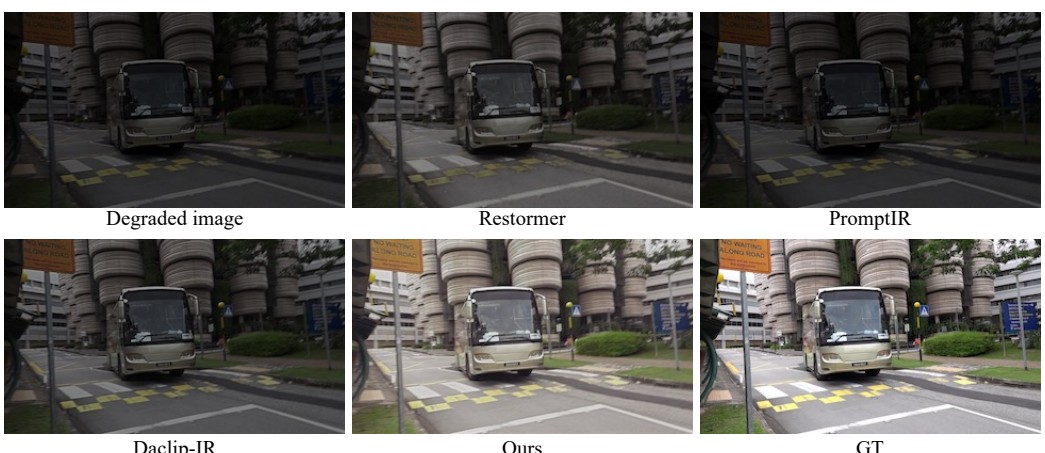

| Degraded image | Restormer | PromptIR |
|---|---|---|
| Daclip-IR | Ours | GT |

Figure 4: Qualitative comparison on multiple degraded images.

## 4.5 Ablation Study

**Complexity Analysis.** We compare model complexity with SOTA models. The comparison results are shown in Table 1, where we report the number of trainable parameters and the runtime for a 256×256 image on an A100 GPU. The complexity of UIR-LoRA is comparable to regression models like NAFNet [2] and significantly more efficient than generative models like Daclip-IR [20].

**Effectiveness of Degradation-Aware Router.** The degradation-aware router plays a crucial role in determining which LoRAs are activated in the inference. To comprehensively demonstrate the impact of the router, we conduct experiments with different selection strategies. As illustrated in Table 3, we have five strategies: "random" indicates activating a LoRA at random, "average" denotes using average weights to activate all LoRAs, and "Top-1", "Top-2" and "All" correspond to setting $K$ in the router to 1,2, and 10, respectively. From the comparison of these results, we can see that the random and average strategies result in poorer performance while using the strategy based on degradation

Table 3: Impact of strategies in router

| Strategy | Multiple Degradation | | | | Mixed Degradation | | | |
|---|---|---|---|---|---|---|---|---|
| | PSNR↑ | SSIM↑ | LPIPS↓ | FID↓ | PSNR↑ | SSIM↑ | LPIPS↓ | FID↓ |
| Random | 17.52 | 0.617 | 0.388 | 126.48 | 10.35 | 0.323 | 0.577 | 104.84 |
| Average | 17.62 | 0.617 | 0.370 | 129.06 | 9.28 | 0.277 | 0.549 | 106.05 |
| Top-1 | 28.06 | 0.864 | 0.105 | 30.62 | 18.04 | 0.683 | 0.321 | 61.65 |
| Top-2 | 28.05 | 0.864 | 0.105 | 30.60 | 18.16 | 0.690 | 0.318 | 61.55 |
| All | 28.05 | 0.864 | 0.105 | 30.61 | 18.16 | 0.691 | 0.318 | 61.58 |

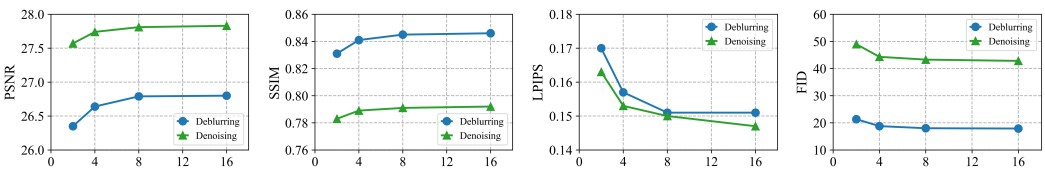

Figure 5: The impact of LoRA's rank on deblurring and denoising tasks.

similarity achieves better outcomes. This suggests that the transferability between different types of degradation is limited and that specific parameters are needed to address their particularities. Furthermore, the selection of the K value also affects the model's performance. When an image has only one type of degradation, a smaller K value can result in comparable performance with lower inference costs. However, for mixed degradations, a larger K value is required to handle the more complex situation.

**Impact of LoRA's Rank.** Within our framework, LoRA is utilized to facilitate the transfer from the pre-trained generative model to the image restoration task. In order to investigate the impact of LoRA's rank on the performance of image restoration, we conduct experiments using deblurring and denoising tasks chosen from ten distinct degradation categories. We set the initial rank to 2 and incrementally increase the value by a factor of 2. The performance changes are depicted in Figure 5. It is evident that as the rank grows, the restoration results improve in distortion and perceptual quality, and at the same time, the number of trainable parameters also increases. Once the rank value exceeds 4, the performance improvement becomes progressively marginal. Therefore, we set the default rank to 4 in our restorer to balance between performance and complexity.

Table 4: The accuracy of predicted degradation type

|  | PSNR↑ | SSIM ↑ | LPIPS ↓ | FID ↓ | Accuracy ↑ |
|---|---|---|---|---|---|
| Original | 26.66 | 0.839 | 0.159 | 18.72 | 91.6 |
| Modified | 26.87 | 0.842 | 0.155 | 18.42 | 99.2 |

**Impact of Predicted Degradation.**

The resizing operation on input images in CLIP models [20, 35] may lead to inaccurate predictions of degradation types, especially for blurry images. To reduce its negative impact on performance, we introduce a simple way that uses the degradation vector of the image crop without resizing to correct the potential error in the resized image. Table 4 is the comparison conducted on blurry images from GoPro dataset. It can be observed that our model with modified operation has higher accuracy and better performance for deblurring.

## 5 Conclusion

In this paper, we propose a universal image restoration framework based on multiple low-rank adaptation, named UIR-LoRA, from the perspective of multi-domain transfer learning. UIR-LoRA utilizes a pre-trained generative model as the frozen base model and transfers its abundant image priors to different image restoration tasks using the LoRA technique. Moreover, we introduce a LoRAs' composition strategy based on the degradation similarity that allows UIR-LoRA applicable for multiple and mixed degradations in the wild. Extensive experiments on universal image restoration tasks demonstrate the effectiveness and better generalization capability of our proposed UIR-LoRA.

## 6 Limitation and Discussion

Although our UIR-LoRA has achieved remarkable performance in image restoration tasks under both multiple and mixed degradations, it still has limitations and problems for further exploration. For instance, adding new trainable parameters into the network for unseen degradations is unavoidable in image restoration tasks, although UIR-LoRA is already more efficient and flexible compared to other approaches.

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

# A  Appendix

## A.1  More Details about Datasets

For multiple degradations, we follow Daclip-IR [20] to construct the dataset, which includes a total of ten distinct degradation types: blurry, hazy, JPEG-compression, low-light, noisy, raindrop, rainy, shadowed, snowy, and inpainting. The data sources and data splits for each degradation type are illustrated in Table 5.

Table 5: Details of the datasets with ten different image degradation types

| Dataset | Train | | Test | |
| | Sources | Num | Sources | Num |
| --- | --- | --- | --- | --- |
| Blurry | GoPro[26] | 2 103 | GoPro | 1 111 |
| Hazy | RESIDE-6k[33] | 6 000 | RESIDE-6k | 1 000 |
| JPEG | DIV2K[1] and Flickr2K[43] | 3 550 | LIVE1[42] | 29 |
| Low-light | LOL[46] | 485 | LOL | 15 |
| Noisy | DIV2K and Flickr2K | 3 550 | CBSD68[23] | 68 |
| Raindrop | RainDrop[32] | 861 | RainDrop | 58 |
| Rainy | Rain100H[50] | 1 800 | Rain100H | 100 |
| Shadowed | SRD[34] | 2 680 | SRD | 408 |
| Snowy | Snow100K-L[18] | 1 872 | Snow100K-L | 601 |
| Inpainting | CelebaHQ[12] | 29 900 | CelebaHQ and RePaint[19] | 100 |

For mixed degradations, we utilize images from REDS[25] and LOLBlur[58]to evaluate the transferability of models. We sample 60 images from REDS and 200 images from LOLBlur dataset for testing. The degraded images from REDS dataset feature a variety of realistic scenes and objects, which suffer from both motion blurs and compression. And the images from LOLBlur dataset cover a range of real-world dynamic dark scenarios with mixed degradation of low light and blurs.

## A.2  More Visual Results

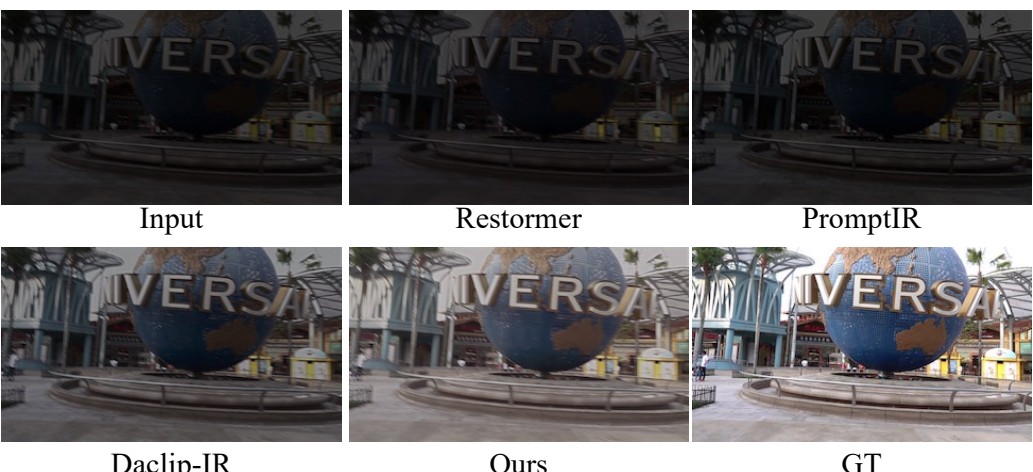

Figure 6: Qualitative comparison on mixed degraded images from LOLBlur dataset.

## A.3  Details about Metrics on Multiple Dagradation

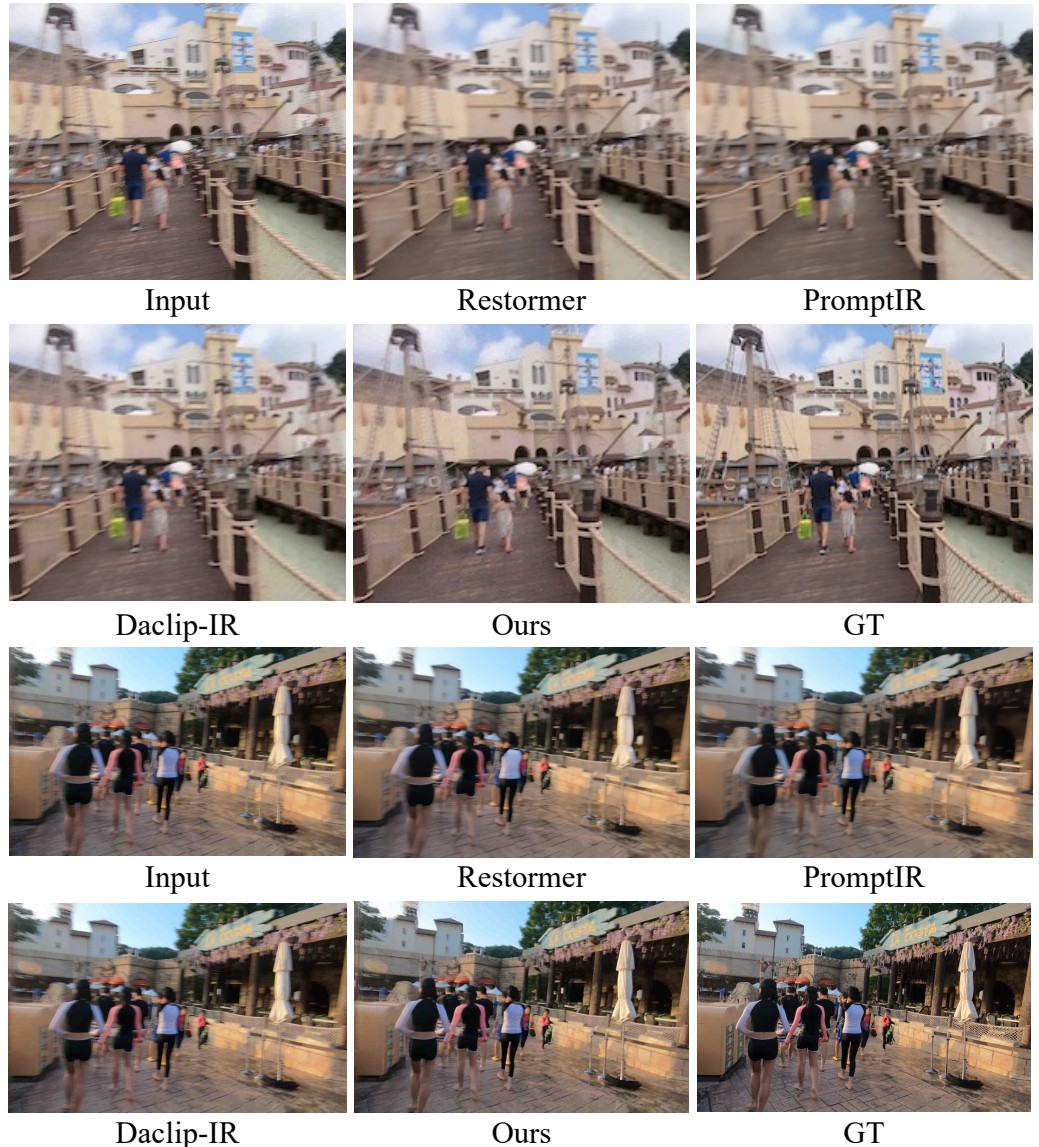

Figure 7: Qualitative comparison on mixed degraded images from REDS dataset.

Table 6: Comparison of the restoration results over ten different datasets on *PSNR*

|  | Blurry | Hazy | JPEG | Low-light | Noisy | Raindrop | Rainy | Shadowed | Snowy | Inpainting | Average |
|---|---|---|---|---|---|---|---|---|---|---|---|
| SwinIR | 24.49 | 23.49 | 24.44 | 19.59 | 25.13 | 24.64 | 22.07 | 23.97 | 21.86 | 24.05 | 23.37 |
| NAFNet | 26.12 | 24.05 | 26.81 | 22.16 | 27.16 | 30.67 | 27.32 | 24.16 | 25.94 | 29.03 | 26.34 |
| Restormer | 26.34 | 23.75 | 26.90 | 22.17 | 27.25 | 30.85 | 27.91 | 23.33 | 25.98 | 29.88 | 26.43 |
| AirNet | 26.25 | 23.56 | 26.98 | 14.24 | 27.51 | 30.68 | 28.45 | 23.48 | 24.87 | 30.15 | 25.62 |
| PromptIR | 26.50 | 25.19 | 26.95 | **23.14** | 27.56 | **31.35** | 29.24 | 24.06 | 27.23 | 30.22 | 27.14 |
| IR-SDE | 24.13 | 17.44 | 24.21 | 16.07 | 24.82 | 28.49 | 26.64 | 22.18 | 24.70 | 27.56 | 23.64 |
| DiffBIR | 22.79 | 20.52 | 22.39 | 16.96 | 21.60 | 23.22 | 21.04 | 22.27 | 20.63 | 18.77 | 21.01 |
| Daclip-IR | **27.03** | 29.53 | 23.70 | 22.09 | 24.36 | 30.81 | **29.41** | 27.27 | 26.83 | 28.94 | 27.01 |
| Ours | 26.66 | **30.28** | **27.15** | 22.45 | **27.74** | 30.51 | 28.26 | **28.63** | **28.09** | **30.88** | **28.06** |

Table 7: Comparison of the restoration results over ten different datasets on *SSIM*

|  | Blurry | Hazy | JPEG | Low-light | Noisy | Raindrop | Rainy | Shadowed | Snowy | Inpainting | Average |
|---|---|---|---|---|---|---|---|---|---|---|---|
| SwinIR | 0.758 | 0.848 | 0.734 | 0.735 | 0.690 | 0.758 | 0.623 | 0.757 | 0.665 | 0.743 | 0.731 |
| NAFNet | 0.804 | 0.926 | 0.780 | 0.809 | 0.768 | 0.924 | 0.848 | 0.839 | 0.869 | 0.901 | 0.847 |
| Restormer | 0.811 | 0.915 | 0.781 | 0.815 | 0.762 | 0.928 | 0.862 | 0.836 | 0.877 | 0.912 | 0.850 |
| AirNet | 0.805 | 0.916 | 0.783 | 0.781 | 0.769 | 0.926 | 0.867 | 0.832 | 0.846 | 0.911 | 0.844 |
| PromptIR | 0.815 | 0.933 | **0.784** | **0.829** | 0.774 | **0.931** | **0.876** | 0.842 | 0.887 | **0.918** | 0.859 |
| IR-SDE | 0.730 | 0.832 | 0.615 | 0.719 | 0.640 | 0.822 | 0.808 | 0.667 | 0.828 | 0.876 | 0.754 |
| DiffBIR | 0.695 | 0.761 | 0.607 | 0.665 | 0.395 | 0.682 | 0.573 | 0.568 | 0.566 | 0.678 | 0.618 |
| Daclip-IR | 0.810 | 0.931 | 0.532 | 0.796 | 0.579 | 0.882 | 0.854 | 0.811 | 0.854 | 0.894 | 0.794 |
| Ours | **0.839** | **0.962** | 0.782 | 0.826 | **0.789** | 0.908 | 0.857 | **0.862** | **0.893** | 0.916 | **0.864** |

Table 8: Comparison of the restoration results over ten different datasets on *LPIPS*

|  | Blurry | Hazy | JPEG | Low-light | Noisy | Raindrop | Rainy | Shadowed | Snowy | Inpainting | Average |
|---|---|---|---|---|---|---|---|---|---|---|---|
| SwinIR | 0.347 | 0.180 | 0.392 | 0.362 | 0.439 | 0.353 | 0.481 | 0.335 | 0.388 | 0.265 | 0.354 |
| NAFNet | 0.284 | 0.043 | 0.303 | 0.158 | 0.216 | 0.082 | 0.180 | 0.138 | 0.096 | 0.085 | 0.159 |
| Restormer | 0.282 | 0.054 | 0.300 | 0.156 | 0.215 | 0.083 | 0.170 | 0.145 | 0.095 | 0.072 | 0.157 |
| AirNet | 0.279 | 0.063 | 0.302 | 0.321 | 0.264 | 0.095 | 0.163 | 0.145 | 0.112 | 0.071 | 0.182 |
| PromptIR | 0.267 | 0.051 | 0.269 | 0.140 | 0.230 | 0.078 | 0.147 | 0.143 | 0.082 | 0.068 | 0.147 |
| IR-SDE | 0.198 | 0.168 | 0.246 | 0.185 | 0.232 | 0.113 | 0.142 | 0.223 | 0.107 | 0.065 | 0.167 |
| DiffBIR | 0.269 | 0.158 | 0.244 | 0.273 | 0.442 | 0.187 | 0.309 | 0.261 | 0.236 | 0.246 | 0.263 |
| Daclip-IR | **0.140** | 0.037 | 0.317 | **0.114** | 0.272 | 0.068 | **0.085** | 0.118 | 0.072 | **0.047** | 0.127 |
| Ours | 0.159 | **0.021** | **0.204** | 0.126 | **0.153** | **0.048** | 0.112 | **0.103** | **0.070** | 0.056 | **0.105** |

Table 9: Comparison of the restoration results over ten different datasets on *FID*

|  | Blurry | Hazy | JPEG | Low-light | Noisy | Raindrop | Rainy | Shadowed | Snowy | Inpainting | Average |
|---|---|---|---|---|---|---|---|---|---|---|---|
| SwinIR | 53.84 | 35.43 | 83.33 | 156.55 | 126.87 | 111.64 | 186.60 | 70.22 | 79.51 | 139.71 | 104.37 |
| NAFNet | 42.99 | 15.73 | 71.88 | 73.94 | 82.08 | 56.43 | 86.35 | 47.32 | 35.76 | 44.32 | 55.68 |
| Restormer | 39.08 | 15.34 | 72.68 | 78.22 | 87.14 | 50.97 | 78.16 | 48.33 | 33.45 | 36.96 | 54.03 |
| AirNet | 41.23 | 21.91 | 78.56 | 154.2 | 93.89 | 52.71 | 72.07 | 64.13 | 64.13 | 32.93 | 64.86 |
| PromptIR | 36.5 | 10.85 | 73.02 | 67.15 | 84.51 | 44.48 | 61.88 | 43.24 | 28.29 | 32.69 | 48.26 |
| IR-SDE | 29.79 | 23.16 | 61.85 | 66.42 | 79.38 | 50.22 | 63.07 | 50.71 | 34.63 | 32.61 | 49.18 |
| DiffBIR | 37.84 | 31.83 | 66.07 | 150.96 | 127.27 | 81.27 | 133.60 | 74.09 | 53.62 | 154.02 | 91.03 |
| Daclip-IR | **14.13** | **5.66** | 42.05 | **52.23** | 64.71 | 38.91 | 52.78 | 25.48 | 27.26 | 25.73 | 34.89 |
| Ours | 18.72 | 5.92 | **37.23** | 62.21 | **44.36** | **23.77** | **44.30** | **23.39** | **22.77** | **23.50** | **30.62** |

Table 10: Impact of rank in LoRAs

| Rank | Deblurring | | | | Denoising | | | |
|---|---|---|---|---|---|---|---|---|
|  | PSNR↑ | SSIM ↑ | LPIPS ↓ | FID ↓ | PSNR↑ | SSIM ↑ | LPIPS ↓ | FID ↓ |
| 2 | 26.35 | 0.831 | 0.170 | 21.35 | 27.57 | 0.783 | 0.163 | 48.98 |
| 4 | 26.64 | 0.841 | 0.157 | 18.79 | 27.74 | 0.789 | 0.153 | 44.32 |
| 8 | 26.79 | 0.845 | 0.151 | 18.01 | 27.81 | 0.791 | 0.150 | 43.29 |
| 16 | 26.80 | 0.846 | 0.151 | 17.90 | 27.83 | 0.792 | 0.147 | 42.82 |

