# OpenReview forum: "UIR-LoRA: Achieving Universal Image Restoration through Multiple Low-Rank Adaptation"
_NeurIPS.cc/2024/Conference — Submitted to NeurIPS 2024_

### Official Review · Reviewer_xL9P · 2024-07-06

**Soundness:** 1
**Presentation:** 2
**Contribution:** 2
**Rating:** 3
**Confidence:** 5

**Summary:**

This paper introduces a universal image restoration framework UIR-LoRA based on multiple low-rank adapters. UIR-LoRA employs the pre-trained text-to-image diffusion model SD-turbo as the shared component. It utilizes a LoRA composing strategy based on the degradation similarity predicted by CLIP encoder to combine different LoRA modules. Experiments show the effectiveness of the proposed method.

**Strengths:**

1. The proposed LoRA-based Universal IR method is easy to understand and follow.
2. The motivation of this paper is very clear to me.

**Weaknesses:**

1. UIR-LoRA adopts SD-turbo as the pre-trained backbone for image restoration. However, SD-tubo utilizes VAE with high compression rate to encode input images, resulting in severe detail distortion for image restoration. This issue has been widely discussed in recent published works [1,2]. However, the paper ignores this very important issue in the Method Section and only mentions the skip-connections for VAE in Line 223.
2. The degradation-aware router seems to be unreliable. I do not believe that the original pre-trained CLIP Text Encoder can distinguish between different degradations through degraded text representations, such as "rain" and "raindrop". Therefore, DA-CLIP fine-tunes the original CLIP. But this paper doesn't contain any discussions about this.
3. This paper does not provide complete technical details, such as how the LQ image is used as a condition for SD-turbo. Is ControlNet used, or is it directly concatenated? I do not see any information about this in the paper.
4. Tab. 1 only reports the trainable Param for UIR-LoRA. I think it's necessary to report the overall Param of the model. In addition, the reported PSNR for DiffBIR is very low. Did the authors add skip-connections to the VAE of DiffBIR for a fair comparison?
5. The visual results in Fig. 3 seem strange. The visual results of Restormer show noticeable artifacts between patches. Do the authors test Restormer using a tiled mode? As far as I know, using a single A100 GPU (Line 251), Restormer can restore the entire image without encountering out-of-memory issues.

[1] Wang, Wenjing, et al. "Zero-Reference Low-Light Enhancement via Physical Quadruple Priors." In CVPR, 2024.

[2] Geng, Zigang, et al. "Instructdiffusion: A generalist modeling interface for vision tasks." In CVPR, 2024.

**Questions:**

1. Authors should discuss the skip-connections for VAE in the Method Section with more details.
2. Can authors provide the degradation prediction accuracy for more different predictions (eg, rain/raindrop)?
3. Authors should provide more technical details of the proposed method.
4. More experimental results and explanations should be included.

**Limitations:**

Yes

---

> ### Author Rebuttal · Authors · 2024-08-07
>
> Thanks for your thorough review and valuable feedback.
> 1. Detail distortion arises from operations such as downsampling or pooling. Using skip connections has become a standard and commonly used method to address this issue, as seen in the bypass decoder in [R1], the skip connections in [R2] and [R3], and even tracing back to the original U-Net. Firstly, this issue is **not our focus** and we have **never claimed** skip-connections **as a contribution** in our paper. Secondly, skip-connections are already **widely adopted** in neural networks and we have provided the references that our paper follows. In these cases, we think our handling of this part is appropriate and will not cause misunderstandings for our method.
> 2. The original CLIP and DA-CLIP do not use text representations to distinguish degradation types during testing; instead, they use image features extracted by the image encoder to classify degradation types. We used the trained DA-CLIP [R4]. The fine-tuning process of CLIP is a contribution of DA-CLIP, and we have never claimed it as our contribution. DA-CLIP uses the extracted degradation representation as input to the attention part of the diffusion process. In contrast, we use the representation to calculate the similarity of each degradation type and then combine the LoRA of different degradations based on this similarity. Compared to DA-CLIP's implicit representation, our way is an explicit and clearly defined approach. Intuitively, explicit methods have stronger interpretability, and experimentally, Sections 4.3 and 4.4 validate the effectiveness of our approach.
> 3. SD-turbo is a one-step method. When using SD-turbo, we did not use ControlNet or concatenated operations; instead, we used the representation of the LQ image in latent space as input directly. This way can be found in [R2]. And we will add this detail in our paper.
> 4. When training and testing DiffBIR, we used its default parameter settings and default structure, which inherently does not include skip-connections. Our experiments were conducted under the same data and environment for comparison. Therefore, there is no issue of unfairness. The lower performance of DiffBIR is due to the need to use SwinIR for preprocessing. When the preprocessing network struggles to handle multiple degradations effectively, the performance metrics of DiffBIR will be lower. The overall parameter can be found in T1.
>
> T1:
> | DA-CLIP  |   SD-turbo  |  LoRAs(trainable)  |   Overall  |
> | :------: | :----: | :-----: | :----: |
> |  125.2M   |  949.1M    |   95.2M             |   1169.5M  |
>
> 5. In the dataset, we test Restormer with the entire image whenever possible. However, for some images, their dimensions cannot be evenly divided by 2 during the forward process. In such cases, we use a tiled mode to test these images, as shown in the JPEG degradation image in Figure 3.
> 6. We used trained DA-CLIP, and except for the blur degradation, the accuracy for the other nine degradations in the test set was 100%, as shown in Table 4 of Daclip-IR. Therefore, in Section 4.5 of our paper, we applied a simple and effective modification to improve the prediction accuracy for motion blur.
>
> R1: Zero-Reference Low-Light Enhancement via Physical Quadruple Priors. CVPR2024.
>
> R2: One-Step Image Translation with Text-to-Image Models. Arxiv2024.
>
> R3: Exploiting Diffusion Prior for Real-World Image Super-Resolution. IJCV2024
>
> R4: Controlling vision-language models for multi-task image restoration.ICLR2024.

---

> > ### Comment · Reviewer_xL9P · 2024-08-13
> >
> > Thank you for your response. I still have some concerns.
> > 1. In Line 146-147, the authors claim "Following Daclip-ir [20], we utilized the pre-trained image encoder in CLIP [35]". This led me to mistakenly believe that the author was using the original OpenAI CLIP.
> > 2. “However, for some images, their dimensions cannot be evenly divided by 2 during the forward process. In such cases, we use a tiled mode to test these images, as shown in the JPEG degradation image in Figure 3.” For low-level vision models, this is a common issue, and the most commonly used operation is to pad the image (e.g., Restormer, NAFNet, Uformer). Why use a tiled mode? When testing your proposed model, how do you handle this kind of situation?

---

> > > ### Author Response · Authors · 2024-08-13
> > >
> > > Thanks for your reply!
> > >
> > > 1. Regarding the use of the pre-trained encoder, we describe it in the "Training and Inference Procedure" section of the paper, specifically on  L195. We apologize for any misunderstanding caused by this sentence in L126. We will revise this sentence to make our description clearer and avoid misunderstandings. Thanks for your suggestions again.
> > >
> > > 2. Since the JPEG-compressed test set contains only 29 images, we checked the size of each image and separately tested those with dimensions not divisible by 2 using a tiled mode in Restormer. Additionally, in the tiled mode, we used the grid function in BasicSR, which is a commonly used method similar to padding.
> > >
> > > We hope our response can address these concerns.

---

> > > > ### Comment · Reviewer_xL9P · 2024-08-14
> > > >
> > > > Thanks for your response. I will consider adjusting the score after the discussion.

---

### Official Review · Reviewer_SkW8 · 2024-07-11

**Soundness:** 2
**Presentation:** 3
**Contribution:** 3
**Rating:** 6
**Confidence:** 4

**Summary:**

This paper proposes to perform universal image restoration via multiple low-rank adaptation. The key idea is to leverage a pre-trained stable diffusion model as the shared component and transfer it to specific degradations with LoRA adaptation. A degradation-aware router is further proposed to generate weights for LoRA combination based on degradation confidence. In experiments, the authors evaluated their method on multi-degradation and mixed-degradation datasets and conducted several ablation experiments on their core components.

**Strengths:**

- The idea of applying LoRA to a pre-trained SD for multi-task image restoration is promising and interesting.
- The overall presentation is easy to follow.
- The experimental results are good and the ablation studies make sense.

**Weaknesses:**

- ControlNet is the most popular approach to adapting SD models to other tasks. I'm curious why the authors chose LoRA? As far as I know, LoRA is often used for large language models (with billions of parameters). It would be great to provide more detailed motivation in the introduction.
- In line 123, maybe it's better to use "concatenate" or other operators instead of "add" to present the unified parameters. Here, the weight $s_k$ can be ignored.
- Can the authors use other SD models as the base model? I believe applying LoRA to a multi-step diffusion process can further illustrate its efficiency.
- In Eq. (4), $s_0 \cdot M_k$ is used in both numerator and denominator, which seems weird and confusing.
- The mixed degradation experiment is cool. It would be interesting if the authors could apply their model to real-world degraded images.
- Line 45: proposed -> propose

**Questions:**

In the degradation-aware router, have you finetuned the CLIP to align degraded images with correct degradation names? How do you choose the degradation names as the vocabulary bank?

**Limitations:**

See Weaknesses.

---

> ### Author Rebuttal · Authors · 2024-08-07
>
> We are truly grateful for your positive feedback on our work.
>
> 1. ControlNet is used in DiffBIR, and it adds a single encoder to handle various degradations, but its performance is still limited by task conflict. However, LoRA can be applied to any layer of a pre-trained model with a small number of parameters and works very well. This is why we chose LoRA in our approach.
> 2. The “+” in L123 corresponds to the parameter merge notation. In practice, we use feature weighting. We will take your suggestion into account and make this notation more accurate.
> 3. Several pre-trained generative models, such as SD1.5, SDXL, SD-turbo, and others, can serve as the base model for our approach. Although using multi-step pre-trained models theoretically offers stronger generative capabilities, we chose the one-step SD-turbo considering efficiency.
> 4. In Eq. (4), we use the letter 'o', not the number '0'.
> 5. In Section 4.4, we used two datasets, REDS and LOLBlur. REDS contains real-world degradations, while LOLBlur, although synthetic, is simulated based on real imaging processes and its data is consistent with real scenes.
> 6.	Thank you for the detailed review of our writing. We will correct this error.
> 7.	In our experiments, we used the trained DA-CLIP. Our vocabulary bank includes ten types of trained degradations.

---

> > ### Comment · Reviewer_SkW8 · 2024-08-13
> >
> > I want to thank the authors for their explanations in the rebuttal. Most of my concerns are addressed thus I will keep my original score.

---

> > > ### Author Response · Authors · 2024-08-13
> > >
> > > Thanks for your reply and positive rating, it means a lot to us.

---

### Official Review · Reviewer_4LsM · 2024-07-12

**Soundness:** 2
**Presentation:** 2
**Contribution:** 2
**Rating:** 5
**Confidence:** 3

**Summary:**

This submission proposes a transfer-learning based strategy to address challenges related to image-degradation restoration. The premise is that a pre-trained generative model can be employed as a common starting component for multiple degradation types, upon which distinct sets of trainable parameters (ie. low-rank adaptors) can be added in order to address specific-degradation restoration tasks. Mixed-degradation restoration is enabled through a top-K hyperparameter, that affords a mixture of (degradation) experts to be active. The experimental setup considers multi and mixed image restoration problems where average results are offered across image-degradation datasets and appropriate standard quantitative metrics, qualitative examples, are reported in comparison with alternative approaches.

**Strengths:**

* The technique described for piping specific samples down specific low-rank adaptor chutes is relatively easy to understand and yet reportedly results in competitive restoration accuracy for investigated datasets.

* Nascent investigations into mixed-degradation image restoration problems provide a promising seed to be followed.

* The writing is of a reasonable standard.

**Weaknesses:**

* The key idea of leveraging pretrained VLM features (and specifically CLIP) for the task of image restoration from multiple degradations, pre-dates the current submission [R1]. While authors clearly go to some length to highlight their alternative CLIP-based scheme, which amounts to envoking specific (pre-existing [R2]) low-rank adaptors, the core technical contributions here can be regarded as somewhat limited.

* The phrase 'Universal Image Restoration' may not be a sufficiently accurate (or modest) description for the proposed method. The submission collates ten different image restoration tasks which, despite vague statements in the abstract, remains a 'multi-task' not a 'universal' setup. Samples for all ten degradation tasks are shared between train and test (Sec. A.1) and individual task adaptors appear to be trained independently on task-specific datasets (L188--196). Generalisation ability to previously unseen degradations is also not considered. Suggest method description requires reworking.

* The claim that multi-task learning (MTL) frameworks, designed to handle image restoration for multiple degradations, share all parameters across different degradations (L029) is incomplete and somewhat misleading. Several existing MTL works (eg. [R3,R4]) make use of both shared and task-specific parameter subsets for multiple image restoration tasks. Indeed 'which proportion of parameters should be shared and which should be task specific' can be considered a fundamental (and long standing) MTL question. The idea of benefiting from commonalities between image restoration tasks is well understood and my concern is that this casts doubt on a core premise of the submission.


References

R1. Controlling Vision-Language Models for Multi-Task Image Restoration. ICLR 2024.

R2. LoRA: Low-rank adaptation of large language models. ICLR 2022.

R3. All in One Bad Weather Removal using Architectural Search. CVPR 2020.

R4. Pre-Trained Image Processing Transformer. CVPR 2021.

Minor:

L076: 'draining' --> 'deraining'

L099: 'mim' --> 'min'

L238: 'aspects' --> 'aspects.'

**Questions:**

> 'for mixed degradations, a larger K value is required to handle the more complex situtation' (L264).

Can additional results be provided for alternative hyperparameter settings (eg. K=1 and K=10) in Tab.2, towards evidencing this claim?

**Limitations:**

Half of one sentence (L293) is apportioned to discussing method limitations. See above for suggestions on components that might make for valid additions here.

---

> ### Author Rebuttal · Authors · 2024-08-07
>
> Thanks for your thorough review and valuable feedback.
> 1. Core technical contributions: The core idea of our method is to introduce the paradigm of **multi-domain transfer learning** into multi-task image restoration, which aims to address the issues of task conflict and feature sharing in multi-task image restoration. This paradigm has not yet been applied in the field of image restoration. Along this line, we focused more on the overall framework rather than technical differences because we believe that the multi-domain transfer learning perspective presented in this paper can inspire more multi-task image restoration methods in the future.
> 2. Method description: Many papers in this field use terms like 'all-in-one'[R1, R2] and 'universal'[R3]. Following these works, we also used the term 'universal.' As you mentioned, multi-task image restoration is more accurate and we will revise this in our paper.
> 3. Claim in introduction: Thank you for your suggestion. We will revise the claim in L29 about 'sharing all parameters.' Our intention was to describe how inappropriate parameter sharing can be one of the sources of task conflict, without affecting the validity of the problem itself or our method. Essentially, our method also addresses how to allocate shared versus task-specific parameters. However, the difference is that we provide such an allocation scheme from the perspective of multi-domain transfer learning.
> 4. Thank you for the detailed review of our writing. We will correct these errors and thoroughly check our paper to ensure there are no other mistakes.
> 5. Additional results: The “mixed degradation” column in Table 3 uses the LOLBlur data from Table 2. Each image in LOLBlur contains at least two types of degradation, so the results for K=2 (Top-2) and K=10 (All) are better than K=1 (Top-1), as shown in the “mixed degradation column” in Table 3. Additionally, we synthesized more mixed degradation data as the reviewer nZDS's request. The impact of the hyperparameter K on the results is shown in T1. When the input image has mixed degradations, a larger K will result in better restoration performance.
>
> T1:
>
> |  Hazy-Blurry-Boisy   |  PSNR  |  SSIM  |  LPIPS  |  FID  |
> | ----- | -------- | -------- | --------- | ------- |
> |  Top-1  |  15.24  |  0.370  |  0.825  |  230.51  |
> |  Top-2 |  15.31  |  0.373  |  0.818  |  230.26  |
> |  Top-3 |  **15.33**|  **0.374** |  **0.817** |  **230.16**|
> |  All   |  **15.33** |  **0.374** |  **0.817**  |  230.17  |
>
> R1. All-in-one image restoration for unknown corruption. CVPR2022.
>
> R2. Promptir: Prompting for all-in-one image restoration.NeurIPS2023.
>
> R3. Selective Hourglass Mapping for Universal Image Restoration Based on Diffusion Model.CVPR2024.

---

> ### Comment · Reviewer_4LsM · 2024-08-12
> **Official Comment by Reviewer 4LsM**
>
> I thank the authors for the response and address their reply point-by-point.
>
> 1. The author rebuttal states that the core technical contributions did not focus on 'technical differences'. For the reviewer, this remains a confusing strategy for the focus of technical contributions. The crux of my concern is that task conflicts and feature sharing are well understood concepts and the rebranding of method terminology ('multi-domain transfer learning') has not added significant additional problem insight, for me.
>
> 2. The authors seem to concede that 'multi-task image restoration' is an accurate description of the work. This appears to further blur any distinction on method terminology.
>
> 3. Assuming the introduction claim is appropriately amended, I can consider this concern well addressed.
>
> 4. I can consider this concern well addressed.
>
> 5. I can consider this concern somewhat addressed and appreciate the additional experimental work. The effect of K in mixed degradation scenarios appears fairly underwhelming (ie. are we near the noise level?). Further I remain unconvinced that PSNRs of circa 12--15 on unknown degradations ('smoky') and mixed scenarios ('Hazy-Blurry-Noisy') are evidencing what might be considered succesful and pragmatic method performance.
>
> In sum, authors address a subset of my concerns however core issues remain problematic. I therefore do not increase my score.

---

> ### Author Response · Authors · 2024-08-13
>
> Thanks for the response from the reviewer. But we cannot agree with the reviewer's response.
>
> 1. Core contribution: We respectfully disagree with the reviewer's view that this is a rebranding of method terminology.
> We introduced the "multi-domain transfer learning" paradigm primarily because pre-trained generative models inherently have **shared image priors across different image restoration tasks**, which current "all-in-one" or "multi-task" image restoration methods have neither proposed nor utilized. Additionally, using parameter-efficient adapters to capture the differences between various degradation restoration tasks and avoid task conflicts is also a novel approach, which has not been explored in the image restoration field.  If the reviewer considers our method to be a "rebranding of method terminology," please provide the references on which this innovative criticism is based. Otherwise, this claim is not convincing.
>
> 2. Method description: "Multi-task image restoration" is a more appropriate term; however, aside from Daclip-IR, other papers with the same setting currently use the terms "all-in-one" or "universal." We have simply followed the common usage but are also considering adopting the more accurate term "multi-task image restoration," as suggested by the reviewer. The reviewer claims that we only trained on 10 types of degradation, but we also tested on mixed degradation and included experiments with unknown degradation during the rebuttal period. Whether in terms of the types of degradation or the complexity of degradation, we have already surpassed all-in-one methods like AirNet and PromptLR. We have covered the diversity of image restoration tasks as comprehensively as possible. From this perspective, why should it not be considered "universal"?
>
> 5. Additional results: First, please do not overlook the scenarios in which 𝐾 is used in our method. It addresses mixed degradation, which involves data distributions that were not encountered during training, and this is inherently a challenging problem. We are unsure where the reviewer’s term "near the noise level" comes from. Could you please provide more specific details? If the reviewer is referring to the data we tested, please note that our method shows improvements across four metrics and has been tested on multiple images, which avoids fluctuations based on a single image or a specific metric. Additionally, the data used in the "smoky" and "Hazy-Blurry-Noisy" experiments are also out-of-distribution. Existing image restoration methods generally struggle with this case, and even many methods have not considered such a case. Please also do not overlook the performance improvements of our method compared to other SOTA methods.

---

> ### Comment · Reviewer_4LsM · 2024-08-13
>
> I thank the authors for their further thoughts and the discussion.
>
> 1. On core contribution - I appreciate the authors' reply around using shared image priors across different image restoration tasks. Apologies if the use of the term "rebranding" was clumsy, I intend to communicate that any distinction with a core strategy of augmenting a pre-trained model with task-specific components remains somewhat subtle for this reviewer. Multi-task methods have indeed made use of 'shared image priors' for some years (e.g. [R1]), for multiple image restoration tasks. I consider the novelty of core-strategy somewhat modest, however the combination of method components do indeed differ.
>
> 2. I agree that authors experimentally cover a diverse set of image restoration tasks. On "universal" terminology; this is largely a matter of taste. If we define the word as "applicable to all cases", then one might expect similar ID / OOD performance. The comment was originally designed to help the authors with reader expectation management. I appreciate that opinions may diverge on this.
>
> 3. My concern here is largely about the statistical power of the effect size relating to K. Additionally; if we are considering PSNR deltas on the order of 10^-2, are these differences qualitatively distinct in image space? meaningful? I find the evidence relating to 'choice of K' claims not overly convincing. I acknowledge that OOD scenarios are challenging (see previous comment) and that method performance improves c.f. alternatives however strong performance, in absolute terms, would seem to evade all methods under comparison.
>
> In sum the authors and I seem unable to reach agreement on a subset of points, however these may not be pathological in nature. I'm happy to somewhat modify my score to reflect this.
>
> References
>
> R1. Chen et al. 2021. Pre-trained image processing transformer. CVPR 2021.

---

> ### Author Response · Authors · 2024-08-13
>
> Thanks for your patient discussion. We welcome the reviewer to discuss our paper based on fairness and objectivity.
>
> 1. Although IPT[R1] has pre-training and fine-tuning stages, it differs significantly from our method. IPT uses synthesized degraded image pairs and learns the mapping from degraded images to clear images during the pre-trained stage. This involves incorporating priors about the degradation process, which is different from the image distribution prior used in the generative models of our method. Additionally, during the fine-tuning stage, the shared parameters are still updated while our pre-trained generative model is frozen. This means that it is necessary to reload the pre-trained parameters in IPT and retrain the parameters when testing different degraded images. Our pre-trained generative model is frozen, meaning the shared parameters have the same weights when testing different degradation types. In contrast, IPT’s shared parameters have different weights during testing. With the fine-tuning stage and the need for explicit degradation categories, this method cannot even be classified as an all-in-one approach. This is why AirNet[R2] claims that they are the first to propose the all-in-one restoration task.
>
> 2. Thanks for the reviewer's suggestions. The core issue with using terms like "multi-task," "all-in-one," or "universal" is how to define the range of degradation types included in the term "all." Currently, this range is not clearly defined in existing papers. Based on the response, the reviewer agrees with this reason. While we believe "multi-task" is a suitable term in this field, the experiments we conducted also support our claim of universal image restoration. And considering the "reader expectation management" mentioned by the reviewer, we will revise this description as we promised in the original rebuttal.
>
> 3. If the reviewer thinks the performance improvement of our 𝐾 strategy is minimal, it will be better if they can refer to the ablation studies of existing restoration methods, such as HInet[R3] and Restormer[R4]. For example, in Restormer, each innovative module has an improvement in PSNR between 0.05 and 0.2. Additionally, the reviewer compares the results of 𝐾=1 and 𝐾=2. But the model with 𝐾=1 is already an improved version of our method, not the "baseline." Our model with 𝐾=1 can handle the primary degradation in the image, and increasing 𝐾 aims to address secondary degradations in case of mixed degradation. We will add a comparison of the processing results when K has different values in the paper if we have the opportunity to submit a camera-ready version.
>
> R1. Pre-trained image processing transformer. CVPR 2021.
>
> R2. All-in-one image restoration for unknown corruption. CVPR2022.
>
> R3. Hinet: Half instance normalization network for image restoration. CVPR2021.
>
> R4. Restormer: Efficient transformer for high-resolution image restoration. CVPR2022.

---

> ### Comment · Reviewer_4LsM · 2024-08-13
>
> I thank the authors for their further clarifications and robust discussion.
>
> 1. The additional dialogue on distinction to previous work, with regard to incorporation-of-priors strategy, is helpful and further alleviates my concerns on this point.
>
> 2. I think we have (somewhat) converged here.
>
> 3. I agree with the authors that qualitative examples will likely further aid reader understanding on method sensitivity to this hyperparameter and would welcome such additions.
>
> As previously noted, my remaining concerns can be considered smaller and I believe my rating, confidence scores now reflect this accurately.

---

> > ### Author Response · Authors · 2024-08-14
> >
> > Thank you for your discussion. Your positive rating is meaningful to us.

---

### Official Review · Reviewer_nZDS · 2024-07-15

**Soundness:** 3
**Presentation:** 3
**Contribution:** 3
**Rating:** 6
**Confidence:** 5

**Summary:**

The paper proposes universal image restoration framework using multiple low-rank adapters that learns task specific weights from to perform multi-domain transfer learning. the proposed method leverages the pre-trained generative model weights as the shared component and adapts it task specific low-rank adapters. At each layer in the restoration pipeline the proposed method uses the degradation similarity to combine LoRA adapters outputs, this enables the proposed to handle for mixed degradation restoration.

**Strengths:**

- The paper proposes LoRA adapters to learn task specific weights and proposes a strategy to combine the adapter outputs using degradation similarity measure
- extensive experiments are performed showing the proposed strategy works better than random and average in table 3.
- extensive experiments are performed to show the proposed methods performance against the sota methods in table 1 for mutliple degradation task.
- Extensive experiments are performaed showing impact of LoRA rank and prediction accuracy

**Weaknesses:**

- In table of the paper authors compared proposed method against sota on REDS and LOLBlur datasets, both these datasets have mixed degradations  of blur, jpeg compression, noise, and low light. Although these comparisons performed on mixed degradations, it would be helpful to how the proposed method performs on mixed weather conditioned images (MID6), which is comparatively challenging than REDS and LOLBlur  datasets.
MID6: Multimodal Prompt Perceiver: Empower Adaptiveness, Generalizability and Fidelity for All-in-One Image Restoration, CVPR, 2024.

- Can authors confirm, whether network re-trained seperately for each experiment in table-1,  and table-2 separately, i.e. table-1 and table-2 trained network weights for proposed method are different.

**Questions:**

- Can the proposed network handle unknown degradation present in the input degradation image
- From the table-3, it is evident that Top-1, and top-2 are almost same performance as All, can authors coment on this, this makes wonder if the input image has only one degradation as dominating for this experiment. Can authors show this experiment on different dataset like MID6

**Limitations:**

- authors have addresed limitations

---

> ### Author Rebuttal · Authors · 2024-08-07
>
> Thanks for your valuable suggestions and for acknowledging our work.
> 1. Why REDS and LOLBlur？ We used the REDS and LOLBlur mixed datasets because the mixed degradation scenarios in these datasets are common, whereas the mixing in MID6 is not commonly seen in real-world scenarios. Since MID6 has not released its data, we simulated the mixing of three types of degradations based on the imaging process and compared our method with several representative image restoration methods. The results are shown in T1.
>
> T1:
> |   Smoky          |  PSNR   |  SSIM  |  LPIPS  |  FID  |
> | ------------- | --------- | ------------- | --------- | ------------- |
> |  Restormer  |  10.54  |  0.418  |  0.604  |  265.99  |
> |  PromptIR   |  10.62  |  0.421  |  0.601  |  263.61  |
> |  Daclip-IR  |  10.97  |  0.420  |  0.564  |  221.98  |
> |  Ours       |  **12.71**  |  **0.476**  |  **0.513**  |  **185.42**  |
>
> 2. Network weight: The network used in Table 2 was tested directly after being trained as shown in Table 1. This means that the networks for testing in Table 1 and Table 2 use exactly the **same parameters and weights**, which is intended to validate the performance of our method on mixed degradation data.
>
> 3. Unknown degradation: The REDS and LOLBlur datasets we used contain mixed degradations that are common in real-world scenarios and are also outside the distribution of the training set. From the results in Table 2, we can see the advantages of our method. Additionally, we tested the performance on the unknown degradation - smoky. The comparison with representative restoration methods is presented in T2, showing that our method has a performance improvement compared to Restormer, PromptIR, and Daclip-IR.
>
> T2:
> |  Hazy-Blurry-Boisy   |  PSNR  |  SSIM  |  LPIPS  |  FID  |
> | ----- | -------- | -------- | --------- | ------- |
> |  Restormer  |  13.39  |  0.074  |  1.312  |  328.97  |
> |  PromptIR   |  13.61  |  0.074  |  1.304  |  330.60  |
> |  Daclip-IR  |  14.50  |  0.181  |  1.028  |  275.37  |
> |  Ours  |  **15.33**|  **0.374** |  **0.817** |  **230.16**|
>
> 4. Table 3 explanation and additional experiment: When the image has only one type of deterioration, the “top-1” strategy and “top-2” strategy perform similarly, as indicated in the “Multiple Degradation” column of Table 3. However, when the degraded image has more than one type of degradations, the “all” strategy and the “top-2” strategy outperform the “top-1” strategy, as illustrated in the “Mixed Degradation” column of Table 3. We conduct the experiment on the data of “Hazy-Blurry-Noisy” like MID6, As shown in T3, since the images have three types of degradations, the “top-2”, “top-3”, and 'all' strategies are superior to the 'top-1' strategy.
>
> T3:
> |  Hazy-Blurry-Boisy   |  PSNR  |  SSIM  |  LPIPS  |  FID  |
> | ----- | -------- | -------- | --------- | ------- |
> |  Ours (Top-1)  |  15.24  |  0.370  |  0.825  |  230.51  |
> |  Ours (Top-2)  |  15.31  |  0.373  |  0.818  |  230.26  |
> |  Ours (Top-3) |  **15.33**|  **0.374** |  **0.817** |  **230.16**|
> |  Ours (All)    |  **15.33** |  **0.374** |  **0.817**  |  230.17  |
>
> R1: MID6: Multimodal Prompt Perceiver: Empower Adaptiveness, Generalizability and Fidelity for All-in-One Image Restoration, CVPR, 2024.

---

> ### Comment · Area_Chair_a8eu · 2024-08-14
> **Reminder for review**
>
> Dear Reviewer nZDS, I have noticed that you have not yet responded to the authors' rebuttal. I kindly urge you to engage in a discussion with the authors at your earliest convenience to help advance the review process.

---

### Official Review · Reviewer_Siin · 2024-07-16

**Soundness:** 4
**Presentation:** 4
**Contribution:** 3
**Rating:** 4
**Confidence:** 5

**Summary:**

This paper presents a framework to improve image restoration across various degradation types using Low-Rank Adapters (LoRA). The proposed method adapts a pre-trained generative model to each degradation type. It performs a weighted sum of the output of adapted models using the estimated degradation of input images. The proposed method performs impressive results in restoration accuracy and resources.

**Strengths:**

The proposed method is interesting and reasonable.
Experimental results support this paper's contributions and the proposed method's effectiveness.

**Weaknesses:**

In Table 3, the 'Top-1' strategy performs almost the same as the 'All' strategy, which limits the motivation of the weighted sum of the adapted models.
Table 6 presents the restoration performance comparisons for each degradation. The proposed method underperforms previous works in significant degradation types such as blurry, low-light, raindrop, and rainy.
The average scores might mislead the evaluation performances.

**Questions:**

How about comparing the proposed method with the following paper?
Selective Hourglass Mapping for Universal Image Restoration Based on Diffusion Model, CVPR 2024

**Limitations:**

The proposed method is simple and effective, but evaluating average scores on multiple degradations can mislead its contribution.
The proposed method achieves near-best performance by selecting a single adapted model but underperforms in many major degradation types.

---

> ### Author Rebuttal · Authors · 2024-08-07
>
> Thanks for your thorough review and valuable feedback.
> 1. Motivation of the weighted sum: When the image has only one type of deterioration, the “top-1” strategy and “all” strategy perform similarly, as indicated in the “Multiple Degradation” column of Table 3. However, when the degraded image has more than one type of degradation, the “all” strategy outperforms the “top-1” strategy. Our motivation for utilizing weighted sum is to enable the model to **perform better in cases of mixed degradation**.
> 2. Why average scores? : **It is difficult to accurately assess the quality of an image using the PSNR metric alone**, as shown in Table 6. Therefore, we also employed **SSIM**, **LPIPS**, and **FID** metrics, as shown in Tables 7, 8, and 9. From the comparison results in Tables 6, 7, 8, and 9, we can see that for the blurry, raindrop, and rainy tasks, our method achieves the best score on at least one or more of the four metrics. Despite the LOL dataset including just 15 images, our metrics are the second best among all the compared methods. Additionally, our focus is on multi-degradation image restoration tasks and average metrics provide a **concise and comprehensive measure of multi-task image restoration models' performance**, which is a common practice in this field, as seen in Table 3 of DA-CLIP.
> 3. Additional comparison: We retrain the method of [R1] and test it using the multiple degradation dataset and the mixed degradation dataset. The results are displayed below. It is evident that our method **outperforms R1** on the four metrics as well as on the average metric.
>
> T1: Multiple Degradation:
> |       |  Blurry  |  Hazy   |  Jpeg   |  Low-light  |  Noisy  |  Raindrop  |  Rainy  |  Shadowed  |  Snowy  |  Inpainting  |  Avg-R1  | Avg-Ours |
> | :--------- | :----------: | :---------: | :---------: | :-------------: | --------- | :------------: | :---------: | :------------: | :---------: | :--------------: | :-----------: | :-----------: |
> |  PSNR   |  26.13   |  27.66  |  27.29  |  19.42      |  27.69  |  30.07     |  27.31  |  26.39     |  27.38  |  18.85       |  25.82    | **28.08** |
> |  SSIM   |  0.821   |  0.937  |  0.792  |  0.822      |  0.771  |  0.913     |  0.828  |  0.850     |  0.884  |  0.769       |  0.838    | **0.864** |
> |  LPIPS  |  0.238   |  0.042  |  0.276  |  0.159      |  0.231  |  0.068     |  0.189  |  0.112     |  0.088  |  0.251       |  0.165    | **0.104** |
> |  FID    |  35.43   |  9.32   |  70.99  |  63.41      |  87.24  |  32.24     |  77.54  |  29.84     |  29.82  |  143.66      |  57.95    | **30.58** |
>
> T2: Mixed Degradation:
> |         |  REDS        |              |              |              |  LOLBlur     |              |              |              |
> | -------- | :--------------: | :--------------: | :--------------: | :--------------: | :--------------: | :--------------: | :--------------: | :--------------: |
> |        |  PSNR        |  SSIM        |  LPIPS       |  FID         |  PSNR        |  SSIM        |  LPIPS       |  FID         |
> |  R1    |  24.93       |  0.710       |  0.413       |  129.17      |  15.77       |  0.612       |  0.380       |  65.70       |
> |  Ours  |  **25.11**  |  **0.718**  |  **0.315**  |  **89.79**  |  **18.16**  |  **0.690**  |  **0.318**  |  **61.55**  |
>
> R1. Selective Hourglass Mapping for Universal Image Restoration Based on Diffusion Model, CVPR 2024

---

> > ### Comment · Reviewer_Siin · 2024-08-13
> > **Thanks for the rebuttal**
> >
> > I appreciate the authors' efforts for the rebuttal. However, the response does not alleviate my concerns. In the case of mixed degradation, the performance difference between the Top-1 and All strategies is only 0.12 dB, and there is little performance difference between the All and Top-2 strategies. Additionally, there is insufficient motivation for the restoration tasks differently from those used in existing universal restoration models. I will maintain my original score.

---

> > > ### Author Response · Authors · 2024-08-13
> > >
> > > We appreciate the reviewer's response, but it is hard for us to agree with your subsequent comments, particularly regarding "those used in existing universal restoration models."
> > >
> > > Based on the reviewer's initial feedback, we have added the additional method for comparison and explained the performance of different strategies such as "top-1," "top-2," and "all." Regarding the new concern of "insufficient motivation" raised by the reviewer, our paper already provides a detailed discussion. We have also further discussed the innovative aspects raised by Reviewer 4Lsm. Please refer to that. If the reviewer believes our method is difficult to distinguish from existing universal restoration models, please provide specific references and indicate which aspects are insufficient. Otherwise, it is not convincing and we can't accept this comment.

---

> > > > ### Comment · Reviewer_Siin · 2024-08-13
> > > >
> > > > It seems some misunderstanding regarding the sentence, "Additionally, there is insufficient motivation for the restoration tasks differently from those used in existing universal restoration models." To clarify, "those" refers to the restoration tasks, not the methods. This connects to my initial review point: "The proposed method underperforms previous works in significant degradation types such as blurry, low-light, raindrop, and rainy. The average scores might mislead the evaluation performances."
> > > >
> > > >
> > > > As I commented, the proposed method is interesting, but the experimental results are limited to support its contributions.

---

> > > > > ### Author Response · Authors · 2024-08-14
> > > > >
> > > > > Thanks for your patient response. We are willing to provide a clearer description to address this issue.
> > > > >
> > > > > First, it is necessary to clarify that using average metrics is one of the common ways in all-in-one image restoration, like Table4 in [R4] and Table3 in [R5]. Additionally, performance evaluation needs to be conducted from multiple perspectives, including the comparison of visual results and comparisons across various metrics. We have provided detailed results for these aspects in our paper.
> > > > > About PSNR in Table6 in our paper, as we mentioned in the initial rebuttal, it is difficult to accurately assess the quality of an image using the PSNR metric alone. The reason can be found in numerous image quality assessment studies[R2, R3]. For example, [R2] believes that "improvement in reconstruction accuracy is not always accompanied by an improvement in visual quality". [R3] notices an "increasing inconsistency between quantitative results and perceptual quality". Therefore, we use multiple metrics to assess image quality in our paper, and please do not overlook the improvements of our method in SSIM, LPIPS, and FID.
> > > > > Additionally, we provide a more direct and detailed comparison of our method and R1 on each type of degradation and each metric, as shown in T1.
> > > > >
> > > > > We hope our explanation and comparison with T1 can address this concern.
> > > > >
> > > > > T1:
> > > > > |       |  Blurry  |  Hazy   |  Jpeg   |  Low-light  |  Noisy  |  Raindrop  |  Rainy  |  Shadowed  |  Snowy  |  Inpainting  |  Average  |
> > > > > | --------- | ---------- | --------- | --------- | ------------- | --------- | ------------ | --------- | ------------ | --------- | -------------- | ----------- |
> > > > > |  PSNR-R1   |  26.13   |  27.66  |  **27.29**  |  19.42      |  27.69  |  30.07     |  27.31  |  26.39     |  27.38  |  18.85       |  25.82    |
> > > > > |  PSNR-Ours   |  **26.66**   |  **30.28**  |  27.15  |  **22.45**      |  **27.74**  |  **30.51**     |  **28.26**  |  **28.63**     |  **28.09**  |  **30.88**       |  **28.06**    |
> > > > > |  SSIM-R1  |  0.821   |  0.937  |  **0.792**  |  0.822      |  0.771  |  **0.913**     |  0.828  |  0.850     |  0.884  |  0.769       |  0.838    |
> > > > > |  SSIM-Ours   |  **0.839**   |  **0.962**  |  0.782  |  **0.826**     | **0.789**  |  0.908     |  **0.857**  |  **0.862**     |  **0.893**  |  **0.916**      |  **0.864**   |
> > > > > |  LPIPS-R1  |  0.238   |  0.042  |  0.276  |  0.159      |  0.231  |  0.068     |  0.189  |  0.112     |  0.088  |  0.251       |  0.165    |
> > > > > |  LPIPS-Ours  |  **0.159**   |  **0.021**  |  **0.204**  |  **0.126**      |  **0.153** |  **0.048**     |  **0.112**  |  **0.103**     |  **0.070**  |  **0.056**       |  **0.105**    |
> > > > > |  FID-R1    |  35.43   |  9.32   |  70.99  |  63.41      |  87.24  |  32.24     |  77.54  |  29.84     |  29.82  |  143.66      |  57.95    |
> > > > > |  FID-Ours    |  **18.72**   |  **5.92**   |  **37.23**  |  **62.21**      |  **44.36**  |  **23.77**     |  **44.30**  |  **23.39**     |  **22.77** |  **23.50**      |  **30.62**    |
> > > > >
> > > > >
> > > > > R1. Selective Hourglass Mapping for Universal Image Restoration Based on Diffusion Model, CVPR 2024.
> > > > >
> > > > > R2. The Perception-Distortion Tradeoff. CVPR2018.
> > > > >
> > > > > R3. PIPAL: a Large-Scale Image Quality Assessment Dataset for Perceptual Image Restoration. ECCV2020.
> > > > >
> > > > > R4. All-in-one image restoration for unknown corruption.CVPR2022.
> > > > >
> > > > > R5. Controlling vision-language models for multi-task image restoration.ICLR2024.

---

### Decision · Program_Chairs · 2024-09-25

**Decision:**

Reject

**Comment:**

The final ratings for this work are as follows: 1 reject, 1 borderline reject, 1 borderline accept, and 2 weak accept, indicating a clear division in the evaluations. Specifically, reviewers nZDS and SkW8 both gave weak accept ratings. Reviewer 4LsM engaged in a thorough discussion with the authors and raised their score to marginal accept, but still expressed concerns about the contribution to performance improvement—a concern shared by Reviewer Siin. Reviewer Siin also had multiple exchanges with the authors and, while his concerns about performance comparisons were somewhat alleviated, he still had reservations about some aspects of the performance improvements. Reviewer xL9P focused more on the technical aspects of the methodology and the detailed explanations of experimental results, ultimately giving a reject rating. Overall, the ratings for this paper are significantly divided. Despite the authors addressing some concerns through in-depth discussions with the reviewers, certain critical issues remain unresolved. Therefore, the paper might still fall short of the NeurIPS 2024 standards, and I am inclined to reject it.